# Perfect Photon Indistinguishability from a Set of Dissipative Quantum Emitters

**DOI:** 10.3390/nano12162800

**Published:** 2022-08-15

**Authors:** Joaquin Guimbao, Lorenzo Sanchis, Lukas M. Weituschat, Jose M. Llorens, Pablo A. Postigo

**Affiliations:** 1Instituto de Micro y Nanotecnología, INM-CNM, CSIC (CEI UAM+CSIC), Isaac Newton 8, Tres Cantos, E-28760 Madrid, Spain; 2The Institute of Optics, University of Rochester, Rochester, NY 14627, USA

**Keywords:** single-photon, quantum optics, photonic integrated circuits, quantum decoherence

## Abstract

Single photon sources (SPS) based on semiconductor quantum dot (QD) platforms are restricted to low temperature (T) operation due to the presence of strong dephasing processes. Although the integration of QD in optical cavities provides an enhancement of its emission properties, the technical requirements for maintaining high indistinguishability (*I*) at high T are still beyond the state of the art. Recently, new theoretical approaches have shown promising results by implementing two-dipole-coupled-emitter systems. Here, we propose a platform based on an optimized five-dipole-coupled-emitter system coupled to a cavity which enables perfect *I* at high T. Within our scheme the realization of perfect *I* single photon emission with dissipative QDs is possible using well established photonic platforms. For the optimization procedure we have developed a novel machine-learning approach which provides a significant computational-time reduction for high demanding optimization algorithms. Our strategy opens up interesting possibilities for the optimization of different photonic structures for quantum information applications, such as the reduction of quantum decoherence in clusters of coupled two-level quantum systems.

## 1. Introduction

Over the last decade, milestones achieved in integrated quantum photonics (IQP) have led to promising results. While other quantum technologies (QT) such as ion trapping or superconducting systems were used to demonstrate their first logical operations in the 1990s [1,2], the first functioning IQP gate was only developed in 2008 [3]. Yet, despite its immaturity, IQP has become established in a wide range of proposed schemes: (Quantum Communications) Si-based chip to chip quantum key distribution (QKD) over 43 km dark fiber was demonstrated in 2018 [4]; network operation for distributed quantum computation (i.e., quantum internet) was reported in 2021 [5]; (Quantum Computation) Gaussian boson sampling scheme with 50 photons for specific quantum computing demonstrated quantum advantage for the first time in 2020 [6]; (Quantum simulation) A IQP based variational eigensolver for calculation of the ground state energy of H_2_ molecules was developed in 2013 [7]; (Quantum Metrology) A IQP platform based on defects in diamond was used for extremely efficient detectors of magnetic fields with unprecedent sensitivity [8]. In contrast to other QT platforms, IQP leverage commercially available systems from the integrated photonics industry, which provide reliable devices for on-chip integration [9] and metamaterial systems for broadband operation [10,11]. In this context, IQP shows a new leading candidate for the future q-bit in QT: the indistinguishable single photon.

Integrated photonics offer different approaches for the modulation of photon emission [12,13]. Suitable platforms for indistinguishable SPS are epitaxially grown semiconductor QDs. QDs enable site control during growth [14] and the possibility of monolithic integration into photonic nanocavities [15,16], providing enhanced quantum emission. As a result, many recent experimental demonstrations have reported record *I* with cavity-integrated QDs at cryogenic T: g^(2)^(0) = 7 × 10^−3^ and *I* = 0.96 was reported with InAs/GaAs QDs embedded in a micropillar cavity at 4.3 K [17]; g^(2)^(0) = 1.2 × 10^−2^ and *I* = 0.97 with InAs/GaAs QDs integrated in a DBR microcavity at 4.2 K [18]; g^(2)^(0) = 2.8 × 10^−3^ and *I* = 0.99 with InGaAs/GaAs QDs inside DBR micropillars at 4 K [19]. However, for T above the cryogenic regime, QDs are subject to pure dephasing mechanisms which reduce the coherence of the emission [20,21,22]: g^(2)^(0) = 0.47 with InGaAs/GaAs QDs at 120 K [23]; g^(2)^(0) = 0.34 with InAs/InP QDs at 80 K [24]; g^(2)^(0) = 0.48 with GaAs/GaAsP QDs at 160 K [25]. For T > 200 K the best reported value is g^(2)^(0) = 0.34 [26]. As a consequence, *I* is reduced to non-practical values for quantum information tasks: *I* > 0.79 for most quantum information processing schemes and *I* > 0.5 for QKD protocols [27]. In this regard, QDs for SPS operation are restricted to low T. In an attempt to overcome this limitation, a variety of cavity-engineering approaches have been conducted [28,29]. However, several theoretical works [30,31,32] indicate that cavity quality factors (*Q)* above 4 × 10^7^ are required for QDs to function at room T, while, to date, the highest reported *Q* coupled to a quantum emitter is about *Q* = 55,000 [33]. In this regard, the theoretical exploration over new strategies for enhancing *I* in the presence of dephasing processes is especially relevant.

Recently, theoretical studies [34,35,36,37,38] have shown that the enhancement and tunability of single photon emission are possible through interfaces based on two-emitter systems coupled to a cavity mode. In their scheme, tunable bandwidth and Purcell enhancement are achieved by dynamical control of the collective states of the two emitters coupled by dipolar interaction. The results open up interesting possibilities for application in single photon generation for quantum information processing. At the same time, deterministic positioning required for dipole–dipole coupling between emitters has been experimentally demonstrated on several SPS platforms: organic molecules [39], color centers in h-BN [40] and diamond [41], terylene molecules [42] and QDs [43,44,45,46]. The potential applications of these cluster systems for the enhancement of *I* have not been studied neither theoretically nor experimentally. As we will show, the cooperative dynamics of these cluster systems can be exploited to maintain high *I* with arbitrary low *Q* cavities by tuning the energy transfer rates between the emitters.

In this work, we present a theory for estimating *I* in a two-emitter system with strong dephasing coupled to a single-mode cavity. We derived an analytical expression of *I* as a function of the distance between the emitters, cavity decay rate, and pure dephasing rate. The results show how the requirements of the cavity for high *I* change with the strength of the dipolar interaction. Taking the model further, we propose a new interpretation of the *I* value, which allows us to estimate its behavior with larger systems (i.e., systems with more than two emitters). We performed numerical simulations of a system of five dipole-coupled emitters to find the optimal configuration for maximum *I*. For the optimization process, we developed a novel machine-learning (ML) scheme based on a hybrid neural network (NN)-genetic algorithm (GA) to find the position of each emitter to maximize *I*. The optimization procedure provides perfect *I* (i.e., *I* = 1) in arbitrary low *Q* cavities, offering unprecedent advantages for relaxing the cavity requirements and favoring the use of QDs as SPS at room T.

## 2. Materials and Methods

### 2.1. Dipole-Dipole Coupling Model

After rotating the wave approximation, the Hamiltonian for the two-QE system shown in Figure 1a coupled to the single-mode cavity reads [35]:(1)$$H=\;\Omega_{12}\left(\sigma_{1}^{†}\sigma_{2}+\sigma_{1}\sigma_{2}^{†}\right)+ig\left(a^{†}\left(\sigma_{1}+\sigma_{2}\right)-a\left(\sigma_{1}^{†}+\sigma_{2}^{†}\right)\right),$$
where $\sigma_{i}/\sigma_{i}^{†}$ are the lowering/rising operators of the QEs and $a/a^{†}$ the annihilation/creation operators of the cavity field. The terms associated with $\gamma ,\gamma^{*}$ and κ are described under Born–Markov approximation, so the evolution of the density matrix follows the Lindblad equation [34]:(2)$$\frac{\partial \rho }{\partial t}=\;-i\left[H,\rho \right]+\;\underset{n}{\sum }(D_{n}\rho D_{n}{}^{†}-\frac{1}{2}\left(D_{n}{}^{†}D_{n}\rho +\rho D_{n}{}^{†}D_{n}\right))\;+2\gamma \underset{i\neq j}{\sum }(\sigma_{i}\rho \sigma_{j}^{†}-\frac{1}{2}\left(\sigma_{j}^{†}\sigma_{i}\rho +\rho \sigma_{j}^{†}\sigma_{i}\right))\;$$
where the $D_{n}$ denotes the collapse operators: $\sqrt{\kappa }a$, and $\sqrt{\gamma^{*}}\sigma_{i}{}^{†}\sigma_{i}$. We have assumed kd≪1 so the modified radiative decay rate is 2 γ and $\Omega_{12}=\;\frac{3\gamma }{4{\left(kd\right)}^{3}}\;$. Without detuning between the QEs there is no coherent coupling between any state but the {|gg⟩,|+〉} set, so the Hamiltonian and Lindblad equation can be written as:(3)$$H=\;\Omega_{12}|+\rangle \langle +|+i\sqrt{2}g\left(a^{†}\sigma_{+}-a\sigma_{+}^{†}\right)\;\frac{\partial \rho }{\partial t}=-i\left[H,\rho \right]+\sum_{n}(D_{n}\rho D_{n}{}^{†}-\frac{1}{2}\left(D_{n}{}^{†}D_{n}\rho +\rho D_{n}{}^{†}D_{n}\right))$$
where $\sigma_{+}=\frac{\sigma_{1}+\sigma_{2}}{\sqrt{2}}$ and now the $D_{n}$ denotes the collapse operators: $\sqrt{\kappa }\hat{a}$, $\sqrt{2\gamma }\sigma_{+}$ and $\sqrt{\gamma^{*}}\sigma_{+}^{†}\sigma_{+}$ [34]. The equations in (14) corresponds to the evolution of a system with a single effective QE with decay rate 2γ coupled to a single-mode cavity field with $\sqrt{2}$*g*. The degree of *I* is defined as [30]:(4)$$I=\;\frac{\iint_{0}^{\infty }dtd\tau {\left|\langle a^{†}\left(t+\tau \right)a\left(t\right)\rangle \right|}^{2}}{\iint_{0}^{\infty }dtd\tau \langle a^{†}\left(t\right)a\left(t\right)\rangle \langle a^{†}\left(t+\tau \right)a(t+\tau )\rangle }$$

Which can be computed numerically via the quantum regression theorem (QRT). Alternatively, to derivate an explicit formula for *I* we start from the following expressions of the master equation:(5)$$\frac{\partial \rho_{ee}}{\partial t}=ig\left(\rho_{ec}-\rho_{ce}\right)-\gamma \rho_{ee}\;\frac{\partial \rho_{cc}}{\partial t}=ig\left(\rho_{ce}-\rho_{ec}\right)-\kappa \rho_{cc}\;\frac{\partial \rho_{ec}}{\partial t}=ig\left(\rho_{ee}-\rho_{cc}\right)-\left(\frac{\Gamma }{2}+\frac{3\gamma }{4{\left(kd\right)}^{3}}\right)\rho_{ec}$$

In the incoherent regime we can apply adiabatic elimination of the coherences by setting $\frac{\partial \rho_{ec}}{\partial t}$ = 0 [30]. Substituting in (5) we obtain the rate Equation (14) with the corresponding transfer rate *R* shown in (13). We can now obtain the numerator of (8) by calculating the ne-G of the system from the equations of motion:(6)$$i\frac{\partial }{\partial t}{\hat{G}}^{R}\left(\tau \right)=i\delta \left(\tau \right)\hat{I}+\left[\hat{H}-i{\hat{\sum }}^{R}\left(0\right)\right]{\hat{G}}^{R}\left(\tau \right),\;{\hat{\sum }}^{R}=\;\left(\begin{array}{cc} \left(\gamma +\frac{\gamma^{*}}{2}\right) & 0\\ 0 & \kappa /2 \end{array}\right)\;$$
where ${\hat{G}}^{R}\left(\tau \right)$ is the retarded ${\hat{G}}^{R}\left(\tau \right)$ and ${\hat{\sum }}^{R}$ the retarded self-energy. Following a similar procedure as in [30], the numerator in (4) can be substituted by:(7)$${\left|\langle a^{†}\left(t+\tau \right)a\left(t\right)\rangle \right|}^{2}=P_{c}^{2}\left(t\right)e^{-\tau \left(\kappa +4g^{2}\Gamma /(\Gamma^{2}+\frac{\gamma^{2}}{{\left(kd\right)}^{6}}\right))}$$

Solving Equation (14) for $P_{C}$ we can analytically solve (4), which gives the expression shown in (13).

### 2.2. Larger Systems

We first obtain the characteristic polynomial of (14): P(λ)=λ2+(κ+2R+1)λ+(κR+κ+R), where we have set γ=1. Then we set the iterative process λn+1⇌P(λn) (from λ=0) and check the stability in the parameter space (κ,R). Figure 2a shows the κ-parameter space of the stability of P(λ) for a fixed *R*. Black dots in the complex plane correspond to κ values whose iteration stays bound and does not diverge to infinity. White dots correspond to values whose iteration diverges to infinity at a maximum speed. Gradient colors correspond to values whose iteration diverges to infinity at different speeds. Our region of interest is the positive real line κ ϵ ℝ+. In this region the iteration diverges to infinity for all κ. We want to measure the speed of the divergence θ for each κ and *R* (i.e., the number of iterations that takes the process to infinity). A good candidate to characterize this value is the slope of P(λ) at λ=0 (i.e., P′(0)). Since P′(λ) grows monotonically with λ, P′(0) uniquely determines θ. In Figure 2b we show the value of P(λ) (blue line) for specific (κ,R). The arrows indicate consecutive λn values of the iteration process. In order to express θ in the decay rate units (λ units), we draw the tangent line to P(λ) at λ=0 (red line in Figure 2b) and take the cut with the x-axis, which gives P(0)P′(0). With this definition θ reads:(8)θ= P(0)P′(0)= κR+κ+Rκ+2R+1

In order to normalize θ, we need to divide (8) by its maximum value $\theta_{max}$. θ is maximum when κ, *R <<* 1, and therefore from (8) we have that $\theta_{max}=\;\kappa +R$. Then the normalized speed of divergence $\bar{\theta }$ is given by:(9)$$\bar{\theta }=\frac{\theta }{\theta_{max}}=\frac{\gamma +\frac{\kappa R}{\kappa +R}}{\kappa +2R+\gamma }=I$$
which matches the expression for *I* [30]. If we apply the same definition of $\bar{\theta }$ for the cascaded cavity system (Equation (15)) we obtain:(10)$$\bar{\theta }=\frac{\kappa_{1}/2+\frac{\kappa_{2}R_{2}}{2(\kappa_{2}+R_{2})}}{\kappa_{1}/2+\kappa_{2}+\frac{3}{2}R_{2}}=I$$
which again matches the expression for *I* [31] after applying the same approximations. In the same way, for the two-emitter system $\bar{\theta }$ matches the *I* value shown in (13). Note that in general the $\frac{P\left(0\right)}{P^{\prime }\left(0\right)}$ is equal to $\frac{\Delta }{\tau }$, where Δ is the determinant and τ is the trace of the rate equations matrix. Therefore, with this method we are able to obtain the analytic expression of *I* for any system from trivial operations in the rate equations, without the need of calculating the ne-G.

### 2.3. Machine Learning Scheme

The Hamiltonian for the 5-QEs system coupled to a single-mode cavity field can be written as:(11)$$H=\;\underset{i\neq j}{\sum }\Omega_{ij}\left(\sigma_{i}^{†}\sigma_{j}+\sigma_{i}\sigma_{j}^{†}\right)+ig\underset{i\neq j}{\sum }\left(a^{†}\left(\sigma_{i}+\sigma_{j}\right)-a\left(\sigma_{i}^{†}+\sigma_{j}^{†}\right)\right)$$
with i,j = (1, …, 5). The modified radiative decay rates $\gamma_{ij}$ and the dipolar interaction strengths $\Omega_{ij}$ can be obtained from the Green’s tensor of the system leading to [38]:(12)$$\gamma_{ij}=\frac{3}{2}\left\{\sin\left(kd_{ij}\right)/kd_{ij})-2\left(\cos\left(kd_{ij}\right)/kd_{ij}{}^{2}\right)-\sin\left(kd_{ij}\right)/kd_{ij}{}^{2}\right\}\;\Omega_{ij}=\frac{3}{4}\left\{-\cos\left(kd_{ij}\right)/kd_{ij})-2\left(\sin\left(kd_{ij}\right)/kd_{ij}{}^{2}\right)-\cos\left(kd_{ij}\right)/kd_{ij}{}^{2}\right\}$$

The evolution of the density matrix follows the Lindblad Equation (2) substituting γ by $\gamma_{ij}$ and adding the corresponding $\sqrt{\gamma^{*}}\sigma_{i}{}^{†}\sigma_{i}$ operators. For each iteration the value of *I* is calculated by solving (2) numerically and computing (8) by QRT. As in each iteration a 12 × 12 matrix is diagonalized, the total time of each function evaluation can take several minutes. At the same time, a GA optimization may require 10^5^ evaluations of the fitness function. If we directly use QRT for each evaluation, the optimization would require excessive computational times. Instead, in our approach we first generate a data set (ω, *I*) with the results obtained from 2000 iterations. With these data, we train a deep NN which learns to estimate the outcome of I for any possible set of random positions $\vec{\omega }$. Now, each time the GA creates a random vector ω, the evaluation of the fitness function obtains *I* from the estimation of the NN. This way, each evaluation takes just a few seconds. Through the iteration of cross-over and mutation, the GA finds the optimal configuration for maximizing *I* after a certain number of generations. Therefore, with our NN-GA scheme we reduce the number of actual numerical simulations for the dataset by two orders of magnitude.

The NN consists of a sequential layer model implemented in Keras module with the corresponding settings: number of layers = 4; neurons per layer = 200; input-dimension = 10; output dimension = 1; loss = mean square error; Epochs = 200; learning rate = 0.001; Batch size = 100; Number of samples = 2000. After the training with 2000 samples both loss and validation-loss converged to 10^−3^, giving enough accuracy for the estimation of *I* and the optimization model. The genetic algorithm uses decimal representation for the genes, one-point crossover and uniform mutation. The total initial population was set to 5000, the number of parents mattings = 2500, number of weights = 1000. Using these values, we needed over 216 generations to find each optimal geometry.

## 3. Results

### 3.1. Indistinguishability of Dipole Coupled Emitters

We consider a system of two quantum emitters (QE) coupled to a single-mode cavity field. Each QE is described by a two-level-system {|g〉,|e〉} with a decay rate γ and a pure dephasing rate $\gamma^{*}$. The QEs interact with each other by direct dipole–dipole coupling with a strength $\Omega_{12}=\;\frac{3\gamma }{4{\left(kd\right)}^{3}}$, where k is the wave vector of the emission and *d* is the distance between the QEs [34]. The cavity field in the Fock basis {|0〉,|1〉} has a decay rate κ and is coupled to the QEs with a coupling constant *g*. Assuming kd≪1 and no detuning between the QEs this system is equivalent to a single effective QE {|gg〉,|+〉} (e-QE) with a decay rate 2γ [34]. Figure 1a shows a layout of the proposed system where Figure 1a (top) shows the two interacting QEs with γ coupled to the cavity field with *g*, and Figure 1a (bottom) shows the equivalent single effective QE system coupled to the same cavity. Here |+〉 represents the superradiant state $|+\rangle =\frac{|eg\rangle -|ge\rangle }{\sqrt{2}}$. The e-QE is coupled to the cavity field with $\sqrt{2}$*g* and a cavity detuning $\delta =\;\Omega_{12}$ [35]. In Figure 1b,c we report the numerical calculation of *I* for the e-QE as a function of the cavity parameters (*g* and κ) for fixed *d,* γ and $\gamma^{*}=10^{4}\gamma$. Figure 1b shows the region of high *I* in the incoherent regime (i.e., *g*
$\ll \;\kappa +\gamma +\gamma^{*}$) while Figure 1c corresponds to the region in the coherent regime (*g*
$\gg \;\kappa +\gamma +\gamma^{*}$). The plots shows a color map with the indistinguishability of the effective QE versus the normalized parameters of the cavity κ and *g*.

Within the incoherent regime the dynamics can be approximated to a population transfer between the e-QE and the cavity field with an effective transfer rate *R* [30]. From the non-equilibrium Green’s function (ne-G) of the system we obtain (see methods):(13)$$R=\;\frac{4g^{2}\Gamma }{\Gamma^{2}+\frac{\gamma^{2}}{{\left(kd\right)}^{6}}}\;,\;I=\;\frac{\gamma \kappa \left[\Gamma^{3}+\Omega_{12}\right]+\left[4g^{2}\left(\gamma +1\right)+\Omega_{12}\frac{\kappa \gamma }{\Gamma }\right]\cdot \left[\Gamma^{2}+\Omega_{12}\right]}{\left[\Gamma^{2}+\Omega_{12}+8g^{2}\right]\cdot \left[\kappa \Gamma^{2}+\Omega_{12}+4g^{2}\Gamma \right]}$$
where Γ = $\gamma +\gamma^{*}+\kappa$. In this regime the cavity behaves as an effective emitter pumped by the e-QE, and the conditions for high *I* are κ<γ and R<γ [30], as shown in Figure 1b. As the distance between the QEs decreases the *R* of the e-QE reduces, so *I* remains high for higher *g* values. This effect is easily visualized in Figure 1d, where we plot the iso-contours of *I* = 0.9 versus the normalized parameters of the cavity κ and *g* for different values of *d*. Each color region in Figure 1d shows the *I* > 0.9 area for a specific value of *d*, which ranges from $d=6.9\times 10^{-2}\lambda$ to $d=8.5\times 10^{-2}\lambda$. Whereas the maximum *g* for *I* > 0.9 is about *g* = 10 γ when *d* = $8.5\times 10^{-2}\lambda$, this value increases to *g* = 20 γ when *d* = $6.9\times 10^{-2}\lambda$. In other words, the requirement for *Q* (i.e., κ<γ) remains unchanged and the *R*-reduction effect just enables high *I* for higher *g* values, which is not particularly interesting. Therefore, the implementation of the two-QE system does not provide any practical advantages (in terms of Q and *g*) with respect to the single-QE. For the three distances, Figure 1f confirms the excellent agreement for *I* values obtained from Equation (1) and from numerical simulations of the two-QE system (see methods).

In the coherent regime the conclusions are roughly similar. Within the range where *g* is close to the strong coupling condition, the (e-QE)-cavity system is equivalent to an effective emitter [30] with decay rate 2γ+R. Here the condition for high *I* is *R* > $\gamma^{*}$ [30], as shown in Figure 1c. Same as before, reducing *d* decreases *R*, requiring higher *g* for high *I*. Figure 1e shows the same iso-contours as Figure 1d in the coherent regime. The *I* > 0.9 region narrows upwards as *d* decreases due to the same *R* reduction effect. Thus, in the coherent regime the two-QE system impose stronger restrictions than the single QE, since it demands higher *g* values for obtaining high *I.* Therefore, the two-QE interface does not provide any advantage for high *I* in terms of cavity requirements, in the incoherent or coherent regimes. However, an extended exploration over systems with larger number of coupled emitters can be relevant. As we will show next, exploiting the cooperative behavior of optimized systems with more than 2 emitters can provide benefits in terms of *I.*

### 3.2. Larger Systems

We showed before that for a set of interacting two-level quantum systems in the incoherent regime the dynamics are described by a population transfer between the subsystems with effective transfer rates *R*. As an example, for a single QE coupled to a single-mode cavity field the evolution of the system reduces to the following rate equations [30]:(14)$$\left(\begin{array}{c} \dot{P_{QE}}\\ \dot{P_{C}} \end{array}\right)=\left(\begin{array}{cc} -\left(\gamma +R\right) & R\\ R & -\left(\kappa +R\right) \end{array}\right)\left(\begin{array}{c} P_{QE}\\ P_{C} \end{array}\right)$$
where $P_{QE}$ is the population of the QE, $P_{C}$ is the population of the cavity and $R=\frac{4g^{2}}{\Gamma }$. As it is described in the Methods section, *I* is obtained from the solution of (14) via the QRT. Since QRT computation is an iterative process, it may be useful to study the dynamic stability of the characteristic equation of (14) to find any kind of relation with *I*. For this purpose, we have defined the degree of stability ($\bar{\theta }$) by measuring the speed of divergence of the characteristic equation of (14) (see Methods). After some algebra, we have found a direct relationship between $\bar{\theta }$ and *I* (see Equations (9) and (10) in the Methods section). This means that we can derive analytic expressions of *I* for arbitrary large system without having to compute the ne-G. Instead, we obtain *I* from the determinant Δ and the trace τ of (14), which significantly simplifies the problem, especially for more complicated systems (such as the ones with more than two emitters). This finding can be expressed as:(15)$$\bar{\theta }=I=\bar{\frac{\Delta }{\tau }}=\frac{\gamma +\frac{\kappa R}{\kappa +R}}{\kappa +2R+\gamma }$$
where $\bar{\Delta }$ is the normalized determinant (see Methods). In the same way as *I*, if κ increases, $\bar{\theta }$ decays at different rates depending on *R*. The alternative interpretation of *I* shown in Equation (15) provides some hints to find a way of keeping high *I* with higher κ values (i.e., to reduce the *Q* of the cavity). For the case of a single QE-cavity system the decay of $\bar{\theta }$ with κ can be tuned by changing *R*. If we include more QEs (or, in general, more subsystems) we have additional transfer rates that may help even more to reduce the cavity *Q*. The additional transfer rates will show up in the off-diagonal terms of the rate equations, giving additional terms in Δ which can lead to new paths to improve the reduction of $\bar{\theta }$ with κ. This approach can be illustrated with the cascaded-cavities scheme [31]. This system considers a single QE coupled to a cavity which at the same time is coupled to a second cavity. In the incoherent regime the dynamics follows the rate equations [31]:(16)$$\left(\begin{array}{c} \dot{P_{QE}}\\ \dot{P_{C1}}\\ \dot{P_{C2}} \end{array}\right)=\left(\begin{array}{ccc} -\left(\gamma +R_{1}\right) & R_{1} & 0\\ R_{1} & -\left(\kappa_{1}+R_{1}+R_{2}\right) & R_{2}\\ 0 & R_{2} & -\left(\kappa_{2}+R_{2}\right) \end{array}\right)\left(\begin{array}{c} P_{QE}\\ P_{C1}\\ P_{C2} \end{array}\right)$$
where $P_{C1}$ is the population of the first cavity, $P_{C2}$ is the population of the second cavity, $\kappa_{1}$ is the decay rate of the first cavity, $\kappa_{2}$ is the decay rate of the second cavity, $R_{1}$ is the transfer rate between the QE and the first cavity and $R_{2}$. is the transfer rate between the first and second cavity. In this case we have one more degree of freedom ($R_{2}$) than in the single QE-cavity system. Therefore, by adjusting $R_{1}$ and $R_{2}$ we can tune the decay of the stability with κ in a more efficient way. Figure 2c shows a quantitative example of this improvement. The plot shows the indistinguishability versus the normalized second-cavity parameter $\kappa_{2}$ for three different values of normalized first-cavity parameter $g_{1}$ = (green), 2γ (red) and 3γ (yellow). While with the single QE-cavity system *I* decreases below 0.5 for κ=γ, the cascaded-cavities scheme can maintain *I* > 0.5 up to $\kappa_{2}=100\gamma$ when setting the right $R_{1}$ and $R_{2}$ values (i.e., setting the cavity mode volume, *V_eff_*, and *Q*).

Therefore, adding more subsystems (emitters and/or cavities) provides additional paths to maintain the stability and, therefore, relax the cavity requirements for high *I*. Accordingly, we study now the case of a cluster of five QEs coupled to a single-mode cavity field. With this scheme, we have 10 transfer rates (*R_ij_*) that can be tuned by setting the relative distances between the QEs, so we have enough parameters to perform a sufficiently complex optimization. Figure 2d shows a layout of the system where each Bloch-sphere represents the time evolution of each QE*_i_*, and each arrow represents the specific transfer rate between the QE*_i_* and QE_j_. Our aim now is to find the geometrical configuration of the QEs that provides the optimal set of *R_ij_* that keep high *I* for high κ values. This goal involves an optimization task with 10 degrees of freedom, which is a highly non-trivial problem and computationally very time-consuming. Nevertheless, similar optimization problems have been recently solved using machine-learning methods [28,47,48,49,50]. Employing a similar approach, we developed a machine-learning scheme based on a hybrid NN-GA algorithm which is able to solve the optimization problem in very short computational times providing the best geometrical configuration for the emitters.

### 3.3. Machine Learning Optimization

We consider five QEs with $\gamma^{*}$ randomly positioned in a 2D-grid. All of them are coupled to a single-mode cavity field with the same coupling constant *g* and cavity decay rate κ. Each relative distance *d_ij_* (*I, j =* 1, …, 5) between QEs leads to a dipolar interaction strength $\Omega_{ij}$ and modified decay rate $\gamma_{ij}$. Since this scheme requires solving a system of 144 coupled differential equations, we are not able to derive an analytic expression for *I* such as in the two-QE case. Instead, we numerically solve the Lindblad equation of the system and compute *I* via QRT. At each iteration we generate a vector ω with five random positions for the QEs and we calculate *I* via QRT for a fixed *g* and κ. The data set (ω, *I*) is then used to train the NN-GA algorithm which finds the optimal positions for maximum *I* for that *g* and κ. In Figure 3a–e we report the obtained optimal geometries for *g* = γ and κ = 10 γ, 50 γ, 100 γ, 500 γ and 1000 γ*,* respectively. All these geometries provide perfect *I* (*I* = 1) with minimum distances *d_ij_*~0.1 λ, a value compatible with experimental realizations [34,35,36,37,38,39,40,41]. Each geometry leads to the right transfer rates *R_ij_* between the subsystems for keeping the stability at the specific rates *g* and κ. For a fixed geometry, small changes in *g* and κ drastically reduce *I*. This is displayed in Figure 3f, which shows *I* versus normalized *g**/*γ and κ/γ for the optimal geometry obtained for *g* = γ, κ = 10 γ. The plot shows a small “bubble” of high *I* at the (*g/*γ, κ/γ) = (1,10) point, while in the neighbor regions of the bubble *I* reduces to 0. Figure 3a–e also shows the positioning tolerances for each QE for obtaining *I >* 0.9. The tolerances for the accuracy in the position depend on the specific QE and the (*g*, κ) values.

Within our scheme the realization of perfect *I* SPS with strong dissipative QEs is possible using well established photonic platforms. To verify this claim we performed 3D-FDTD simulations [51] of a point source placed at the antinode of a cavity-mode in a standard 2D-hexagonal SiN photonic crystal cavity (PCc). The *V_eff_* and *Q* were obtained from the field profile (see Figure 3g) and frequency analysis of the resonance. For a QE with (γ, $\gamma^{*}$, ω) = (160 MHz, 400 GHz and 400 THz) such as color centers in diamond [52] we obtained (*g*, κ) ≈ (1, 100). The radius and distances between the holes of the PCc were set to 120 nm and 50 nm, respectively, which is compatible with most fabrication techniques [53,54,55]. To highlight the benefits of our strategy we have contrasted the obtained performance with standard single-emitter-cavity systems [30] for different QEs at high T. Diamond color centers, InGaAs QDs, GaAs QDs and single molecules at 300 K has a pure dephasing of 1000 γ, 600 γ, 1450 γ and 10^4^ γ, respectively [20,21,52,56]. Considering the same standard PCc with (*g*, κ) ≈ (1, 100), a single-emitter-cavity system leads to *I*~0.01 for all these emitters, whereas the five-QEs optimized platform provides *I* = 1. For these emitters, obtaining *I* = 1 with a single-emitter-cavity at room T would require at least a cavity with *Q* above 4 × 10^7^, which is beyond the state of the art for most current fabrication technologies.

## 4. Discussion

A key point to evaluate for the experimental realization of our scheme is the nanoscale positioning approach for the deposition of the cluster of QDs. Novel positioning technologies have recently shown positioning accuracy at the nanometer level [57]. A 30 nm positioning accuracy with GaAs QDs has been reported using atomic force microscopy [16]. Confocal micro-photoluminescence can provide 10 nm positioning accuracy also with GaAs QDs as it has been shown in [58]. A 5 nm position accuracy has been achieved recently with Bi-chromatic photoluminescence through a new image analysis software implementation [59]. In situ lithography approaches have also shown promising results improving its position accuracy down to 30 nm [60]. Pick-and-place approaches have shown 38 nm positioning accuracy for Si vacancy centers transference to aluminum nitride waveguides, achieving 98% coupling efficiency [61,62]. Therefore, according to tolerances shown in Figure 3b, for the case of point defects in diamond, using pick-and-place positioning we would have a standard deviation of 38 nm with a target of about 30 nm. This leads to 81% probability of successful deposition for a single QD. Successful deposition of the five QDs in place would have a probability of 32%. An experimental realization should require the fabrication of a large number of devices and checking for suitable candidates one by one. According to this, although our scheme could enable the experimental demonstration of certain quantum phenomena, it is still far from a high-scalable technology.

So far, we have explored the theoretical performance of our scheme considering identical QDs without detuning Δ between the emitters. However, a more realistic analysis involves the evaluation of the effect of mismatching between the emission frequencies of the QDs. With this aim, we have incorporated a statistical detuning distribution to the system of five QDs in the configuration shown in Figure 3b. We consider a normal distribution setting the mean equal to 0 and standard deviation $\sigma_{n}=n\gamma$, as shown in Figure 4a. The Δ of each QD is set randomly according to the normal distribution. We start with the distribution $\sigma_{1}=\gamma$, we set five random Δ for the QDs and compute *I*. Then we reset the random Δ according to the same distribution and compute again *I*, repeating this process 200 times and computing the average of all obtained values of *I*. We obtained the average value of *I* for the 20 different probability distributions $\sigma_{n}=n\gamma$ with n=1…20, as shown in Figure 4b.

As expected, the value of *I* reduces as the standard deviation of the distribution increases. For the distribution $\sigma_{1}=\gamma$ the possible values for the Δ between the QDs range from −5 γ to 5 γ, leading to a negligible reduction of *I*. On the opposite side, with $\sigma_{20}=20\gamma$ the possible values of Δ range from −60 γ to 60 γ, giving a reduction of *I* of about 70%. According to these results, our scheme is able to maintain high *I* > 0.75 for normal distributions of emitters with standard deviation below 5 γ, which includes frequency mismatching between the QEs of about 20γ. Therefore, the proposed system is a relatively robust platform for distributions of non-identical QDs according to recent experimental demonstrations [26].

## 5. Conclusions

We have developed an analytical model for estimation of the indistinguishability with two-QE interfaces with dephasing integrated in optical cavities. The model provides an analytical expression that relates the indistinguishability to the distance between the QEs and the parameters of the cavity. Through an alternative interpretation of the indistinguishability, we can estimate the behavior of systems including more QEs. Finally, we performed a numerical optimization of a five-QE system coupled to a single cavity by a machine learning scheme. The results predict perfect indistinguishability with strong dissipative QEs in arbitrary low *Q* cavities. The proposed method provides a strategy for the realization of a source of perfect indistinguishable single photons at room temperature. The strategy presents significant challenges from the perspective of QD positioning process. Although the required accuracy in positioning may be still far from a real scalable technology it can be suitable for experimental demonstration of single photon operation with high indistinguishability. The ML approach may provide insights for optimizing different photonic structures for quantum information applications, such as the reduction of quantum decoherence in clusters of coupled two-level quantum systems.

## Figures and Tables

**Figure 1 nanomaterials-12-02800-f001:**
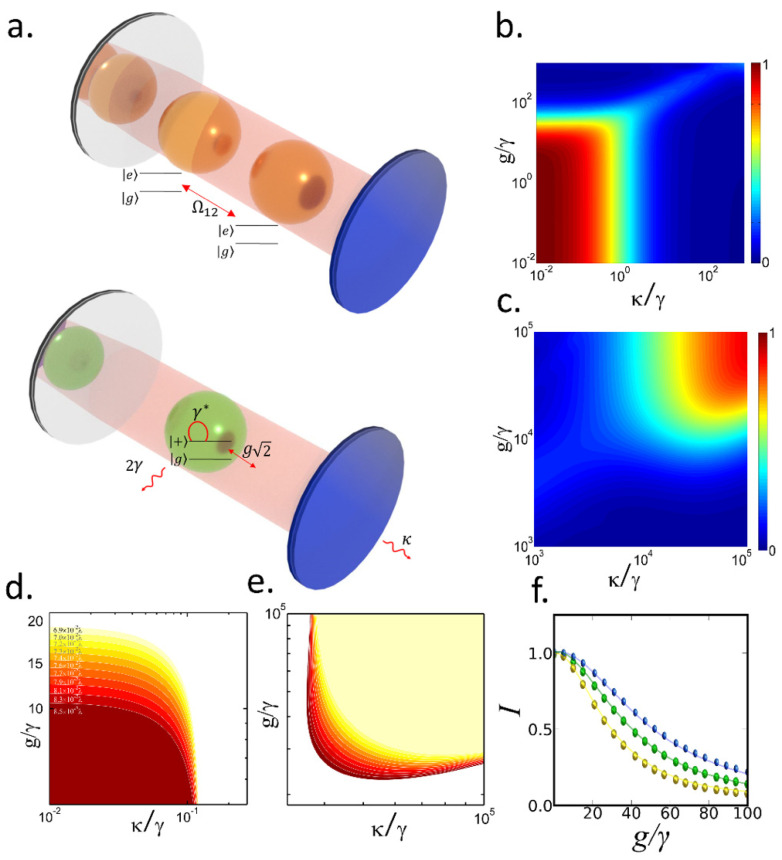
(**a**) The two interacting QEs with γ coupled to the cavity field with *g* are equivalent to a single QE with 2γ coupled to the cavity with $\sqrt{2}$
*g*, each sphere represents a single two-level-system. Indistinguishability of the effective QE versus the normalized κ and *g* in the (**b**) incoherent regime and (**c**) coherent regime. Contour map of regions with *I* > 0.9 for different distances between the emitters from $d=6.9\times 10^{-2}\lambda$ to $d=8.5\times 10^{-2}\lambda$ (**d**) incoherent regime and (**e**) coherent regime. (**f**) Indistinguishability versus normalized *g* for $d=7.2\times 10^{-2}\lambda$ (yellow), $d=7\times 10^{-2}\lambda$ (green) and $d=6.9\times 10^{-2}\lambda$ (blue); solid lines calculated using Equation (1); colored dots obtained from numerical integration of the Lindblad equation with two QEs.

**Figure 2 nanomaterials-12-02800-f002:**
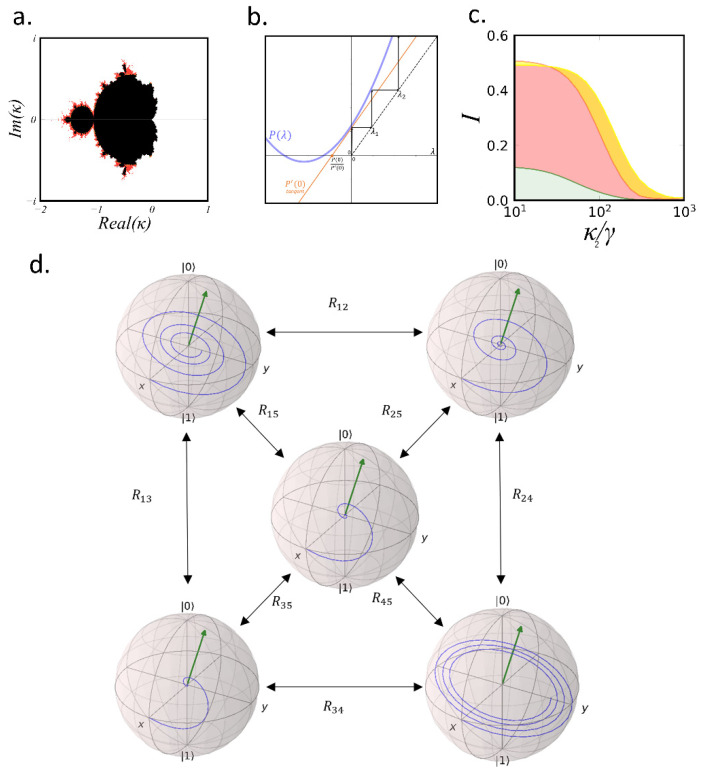
(**a**) κ-parameter space of the stability of rate equations of a single QE system coupled to a cavity. Black dots correspond to bounded points while the gradient colors represent the degree of stability. (**b**) Characteristic equation of (2) (blue line); tangent line with slope $P^{\prime }\left(0\right)$. The cut of the tangent line with the x-axis is given by $\frac{P\left(0\right)}{P^{\prime }\left(0\right)}$. The arrows indicate consecutive $\lambda_{n}$ values of the iteration process. (**c**) Indistinguishability versus normalized $\kappa_{2}$ for $g_{1}$ = (green), 2γ (red) and 3γ (yellow). (**d**) Bloch-spheres of the five-QE system with population rate transfers *R_ij_* between each subsystem.

**Figure 3 nanomaterials-12-02800-f003:**
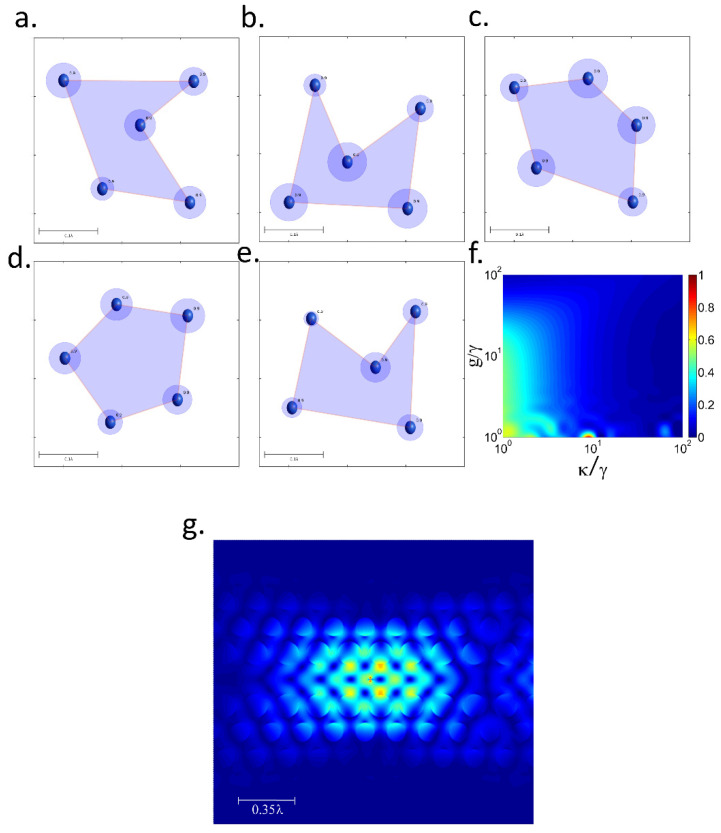
(**a**) Optimal configuration of the 5-QEs system in a 2D plane for (**a**) κ=10γ, (**b**) κ=50γ, (**c**) κ=100γ, (**d**) κ=500γ and (**e**) κ=1000γ. The circles around each QE position corresponds to the positioning tolerance for having *I* > 0.9. (**f**) Indistinguishability versus normalized κ and *g* for the optimized system shown in (**a**). (**g**) Field profile ${\left|E\right|}^{2}$ of the hexagonal PC-cavity-mode with a point source placed at the antinode.

**Figure 4 nanomaterials-12-02800-f004:**
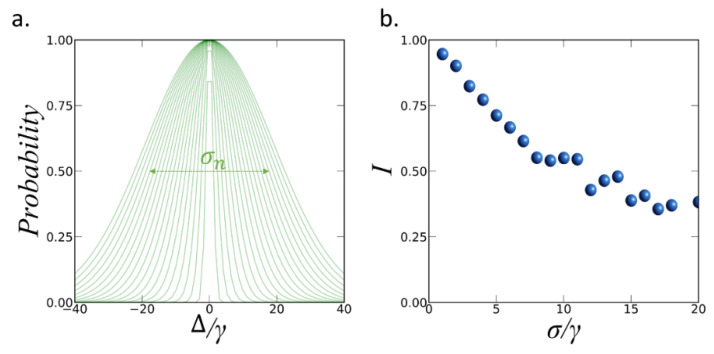
(**a**) Probability distributions with standard deviation $\sigma_{n}$ for *n* = 1 …20 for the normalized detuning values Δ/γ. At each iteration we set a random Δ value for each QD according to the corresponding distribution. (**b**) Average value of the indistinguishability obtained for each of the 20 probability distributions.

## Data Availability

Data underlying the results of this paper are available from the corresponding author upon request.

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
