# Peer review of "Perfect Photon Indistinguishability from a Set of Dissipative Quantum Emitters"

_nanomaterials, 2022, doi:10.3390/nano12162800_

Round 1

Reviewer 1 Report

The manuscript entitled “Perfect photon indistinguishability from a set of dissipative quantum emitters” by Guimbao et al. reported a theory for the estimation of I in a two-emitter system with strong dephasing coupled to a single-mode cavity. They have derived an analytical expression of I as a function of the distance between the emitters, cavity decay rate, and pure dephasing rate. It shows how the requirements of the cavity for high I change with the strength of the dipolar interaction. Furthermore, the authors propose a new interpretation of the I value which allows estimating its behavior with larger systems. The data are solid and convincing, and the manuscript is well written. Therefore, I would like to recommend its publication in Nanomaterials after the authors address the following issues:

1.     By contrasting this strategy with the existing standard procedures, the benefits of this approach are highlighted.

2.     It is preferable for the authors to use concrete examples to demonstrate, as this will be important for future applications.

3.     Some related improvements are made to the manuscript's format. The format mistakes should be checked again. For instance, In the first paragraph of the introduction, line 6 " Quantum Computation [6]6 ", on the third page, the bottom-left sixth line " ? ? ℝ > 0 "

4.     Some of the recent reports on the emitters are worth citing for the benefit of the readers. For example see, Spectrochimica Acta Part A: Molecular and Biomolecular Spectroscopy 217, 86-92; Research doi: 10.34133/2021/8096263.

Reviewer 2 Report

In this paper, the authors developed an analytical model for the estimation of the indistinguishability with two-QE interfaces with dephasing integrated in optical cavities. The model provides an analytical expression which relates the indistinguishability with the distance between the QEs and the parameters of the cavity. Through an alternative interpretation of the indistinguishability, they were able to estimate the behavior of systems including more QEs. Finally, the authors  performed a numerical optimization of a 5-QE system coupled to a single cavity by a machine learning scheme. The results predict perfect indistinguishability with strong dissipative QEs in arbitrary low Q cavities. he  novel points can be seen, and I think this paper can be published after some revision.

1) English in this paper needs improvement, which can make this paper more like a journal paper.

2) Figs.2 and1 is not clear, and the authors can give a more details of those figures.

3) The authors can do a short introduction about the future challenges in the indistinguishable SPST.  For instance, quantum technologies, and algorithm.

4) Please polish the abstract. Please check the logic of abstract. Please add sentences to explain the meaning, the main points, the improvement and the promising application of the study. Plenty of detail data have given, however, in abstract, important procedures and results should be mentioned in simple manner. Please focus on the main points and the improvement of the study..  

5) The authors can do a short introduction about the future development in integrated photonics. Some references can be considered to add in this manuscript. For instance,

#Results in Physics, vol. 12, pp. 917-924, 2018.

Round 2

Reviewer 2 Report

No more commets, and I think this paper can be published.